# The Effectiveness of Family Group Conferencing and the Challenges to Its Implementation: A Scoping Review

**DOI:** 10.3390/nursrep15040122

**Published:** 2025-04-01

**Authors:** Naohiro Hohashi, Qinqiuzi Yi

**Affiliations:** 1Division of Family Health Care Nursing, Department of Nursing, Graduate School of Health Sciences, Kobe University, Kobe 654-0142, Japan; akiko@ctgu.edu.cn; 2Department of Public Administration, School of Law and Public Administration, China Three Gorges University, Yichang 443002, China

**Keywords:** family group conferencing (FGC), family nursing, scoping review, effectiveness, challenges

## Abstract

**Aim:** The aim of this study was to identify the effectiveness of Family Group Conferencing (FGC), a decision-making model that is not only family-centered but also takes the form of a family-driven or social network, and to consider the challenges to FGC implementation. **Methods:** A scoping review was conducted using the Arksey and O’Malley framework. A systematic search was conducted of such electronic databases as PsycInfo, CINAHL, Google Scholar, and Web of Science. Criteria were set utilizing the search terms “family group conferencing” or “family group conference”, with the search refined to studies published between January 2015 and July 2020. The data extracted by the review team were inductively analyzed, and the findings were classified into categories. **Results:** This review included a total of 26 studies. The categories underscoring the effectiveness of FGC included “sense of ownership”, “restoring belongingness”, “reduction of coercion”, and “learning platform”. Categories presenting challenges to FGC implementation included “severe situations of main actor”, “severe situations of the family”, “the complex role of the FGC coordinator”, and “the cost-ineffectiveness of FGC”. **Conclusions:** The effectiveness in the capacity of decision-makers was determined by the interaction between the main actor and social network of the FGC, with the challenges to FGC reducing the likelihood of completing the FGC process. It will be necessary therefore to identify the skills and qualifications of FGC coordinators, who must take into account group dynamics, so as to enable the main actor and their social network to develop a positive reciprocal interaction.

## 1. Introduction

Family Group Conferencing (FGC) is a decision-making model that was developed in New Zealand in the 1980s. In 1989, it was incorporated into that country’s Children, Young Persons, and Their Families Act [1] as an effective method for solving problems involving children. FGC is based on the idea that families themselves, rather than professionals, are the experts on their family’s situation and problems, and postulates that most families have sufficient abilities and resources to make competent decisions concerning their own welfare. This differs, therefore, from the general idea of decision-making.

FGC’s principles and values are based on New Zealand Māori culture. In traditional Māori culture, whānau was the place where family members and extended family members would meet, as well as where initial teaching and socialization took place. Nowadays, whānau has come to mean an environment within which certain responsibilities and obligations are taught to the younger generation [2]. According to Barn and Das (2016) [3], while the roots of the FGC model come from Māori cultural traditions, and the concept of family in FGC was broadly defined to include immediate family members and wider kinship systems, little actual discussion or research on FGC focuses on the system’s cultural and ethnic background.

FGC had been presented at local workshops throughout New Zealand, and it was shared and further developed in Great Britain, as well as in Hawaii and in other US states [4]. FGC studies were conducted across Europe, North America, and Oceania, mostly focused on child welfare and restorative justice [5]. De Jong and Schout (2011) [6] note that studies on the application of FGC in mental health practices are uncommon, but from around 2015, the number of studies using FGC in mental health practices began increasing [7,8,9,10,11,12]. Furthermore, the FGC has been widely implemented in numerous other areas, such as homelessness [13], work disability [14], care for older adults with dementia and their families [15], and so on. FGC has also been evaluated using a variety of strategies, such as qualitative interviews, observations of clients and families, social networks, practitioners, and analysis of administrative data and impact analysis of outcome.

The FGC process consists of five phases: (1) preparation phase, (2) information phase, (3) private time phase, (4) presentation of the plan phase, and (5) carrying out the plan and evaluation phase [16,17]. In the preparation phase, the coordinator helps the main actor to explore his/her social network, and the main actor decides who to invite for the conference. In the information phase, the meeting starts, and professionals can provide the information needed to help answer the main actor’s questions. In the private time phase, the main actor and his/her social network deliberate together, but the professionals and the coordinator are not present. In the presentation of the plan phase, the participants present their plan to the coordinator and the professionals involved. Having professionals involved with families at this stage is different from the general method of decision-making. In the final phase, all those involved carry out the plan and evaluate its effectiveness and results, and then they adjust the plan accordingly.

FGC might have numerous participants, including patients, children, their families, members of their social network, professionals (caseworkers, nurses, and so on), and FGC coordinators. In the field of public mental health nursing, the person suffering from or causing the problem(s) in question is referred to as the main actor, and family or people close to them are referred to as their social network. In the field of FGC for older adults, those who are suffering from health or financial problems, such as caring for a spouse who had suffered a stroke, are referred to as cases and their family (siblings, children, and so on) are referred to as their social network. In this study, “main actor(s)” refers to the primary party(s) in FGC, while “social network” refers to their family, relatives, and friends.

The theory underlying FGC is that people have the right to make their own decisions, and that the main actor and their social network bear the primary responsibility for that person’s problems and for finding solutions to these problems [16]. In this way, FGC is not only family-centered but also family-driven (or social network-driven). The key concepts of FGC are resilience and relational autonomy [18], which are central to the FGC process of interaction between the main actor and social network. Resilience is the ability to adapt to difficult or threatening situations, and it includes the ability to adapt and develop through social and environmental interactions [18,19]. Relational autonomy is defined as an approach to autonomy that emphasizes interaction and growth through one’s relationships [20,21].

Regarding the interaction between the main actor and the social network in FGC, the aim of the interaction is to optimize family decision-making, as it is a process with alternating phases of sharing knowledge and skills, coaching, shared decision-making (SDM), and, eventually, of fully empowering the participants [22]. Metze et al. (2015) [16] suggest that the main actor’s interaction during FGC with a social network helps them validate and gain respect for their own decisions and actions, leading to a changed perspective, while building self-respect and self-esteem, and creating a stronger sense of self-worth.

The SDM model, typically seen in the medical field, also emphasizes the interaction of people in a manner similar to FGC. SDM supports patient/family autonomy through informed consent, and through communication among clinicians, patients, and family members. SDM is defined as an approach where clinicians and patients communicate using the best available evidence when faced with a task [23]. Decision-making, in this view, is important for people’s own values and identity, as well as for the amount of communication between clinicians and patients it provides. Hence, healthcare providers play a key role when informed consent is involved, promoting patients’ autonomy through relational approaches. Specifically, in clinical nursing practice, FGC provides nurses with an effective framework to foster patient and family engagement; enhance communication among care teams; and support holistic, patient-centered care through empowerment and relational autonomy. Moreover, nurses, who typically work closely with patients and their families, are ideally positioned to facilitate FGC, thereby improving care coordination, adherence to treatment plans, and overall health outcomes by leveraging families’ inherent strengths and social networks.

While interactions between people in decision-making support are important, the effects on the lives of patients and their families through those decisions have not been adequately examined. FGC resulted in improvements in the quality of social support, resilience, and living conditions [7]. However, a meta-analysis examining the effectiveness of FGC in youth care indicated FGC to be no more effective than regular care [24]. The mechanisms of FGC are complex and difficult to quantify. Arguments have been raised that randomized trials are not the most appropriate research methods to examine the efficacy of FGC, and that it would be more effective to employ a variety of both qualitative and quantitative research methods to analyze the FGC process [25,26]. It is also believed that, in order to identify the effectiveness of FGC and the challenges to its implementation, it is important to consider the process from multiple perspectives, using qualitative and quantitative research, and that scoping reviews are useful for summarizing research findings and identifying multiple research gaps. As research into FGC progresses, new methods of family decision-making may be developed.

The purpose of this study was to identify the efficacy of FGC and the challenges it faces regarding its implementation by addressing two questions: (1) In what ways has FGC shown to have been effective? (2) What are the key challenges that FGC faces?

## 2. Methods

### 2.1. Study Design

Scoping reviews are useful for summarizing research findings and identifying research gaps [27]. Because the purpose of this study was to identify the areas in which FGC is effective and examine the challenges faced in implementing FGC, a scoping review was employed.

A scoping review was conducted using the method steps outlined in Arksey and O’Malley (2005) [27] and reported in accordance with the Preferred Reporting Items for Systematic Review and Meta-Analysis Extension for Scoping Reviews (PRISMA-ScR) [28]. As this is the first systematic review on this topic, the review protocol has not been registered.

### 2.2. Definition of FGC

FGC is defined as “an intervention in which a plan is not made by a professional, but by the person who needs help and support himself/herself, together with his/her social network” [18].

### 2.3. Information Sources and Strategies

A systematic search was conducted through the electronic databases in PsycInfo, CINAHL (Cumulative Index to Nursing and Allied Health Literature), Google Scholar, and Web of Science. The search keywords were set to “family group conferencing” or “family group conference”. The range of publication was set from January 2015 to July 2020, the reason being that 2015 marks the point from which FGC began to be widely implemented in numerous fields and in various forms.

### 2.4. Exclusion Criteria

Publication exclusion criteria were articles not published in English and articles that made no reference to any kind of intervention process, outcome, and effect or problem of FGC. This review did not assess the methodological quality of the included studies.

### 2.5. Charting the Data

A descriptive–analytical method was used for data extraction from the included studies. The following information was extracted: study characteristics (author, date of publication, country, aims, and study design); participant demographics; study settings; description of the FGC intervention (e.g., coordinator role and length of time for FGC intervention); outcomes and outcome measures; and key findings, including the efficacy and challenges of FGC intervention.

An inductive thematic approach was used for identifying and coding contributing factors. Through this analysis, the areas in which FGC is effective and the challenges faced in implementing FGC were extracted.

Descriptive data from individual studies were collated, and the effectiveness and challenges to FGC were identified for each context. Next, each description was divided into codes, which are small meaning units. Code classification was performed by focusing on similarities and differences, with codes found to be conceptually similar being grouped into more abstract concepts, termed subcategories. The subcategories were then grouped to create categories, and their abstraction levels were refined and checked by the review team. Specifically, the trustworthiness of all analyses underwent independent analysis by two family health nursing researchers and was reviewed until a consensus was reached. If there was no consensus between the two researchers, both researchers re-evaluated the paper. If no consensus was reached between the two researchers, opinions were exchanged repeatedly with 11 collaborating researchers until a final decision was reached.

## 3. Results

### 3.1. Study Selection

The flow of study identification and selection is shown in Figure 1. The search identified 161 articles: PsycInfo (*n* = 35); CINAHL (*n* = 25), Google Scholar (*n* = 50), and Web of Science (*n* = 51). After removing 101 duplicate articles, 60 abstracts were screened, and then those papers which were abstract inaccessible (*n* = 6), not available in English (*n* = 1), and not reporting FGC (*n* = 17) were excluded. Thirty full papers were accessed for further evaluation based on the exclusion criteria (no reference to any kind of intervention process, outcome, and effect or problem of FGC), resulting in the final review of 26 full-text articles (43.3% of identified citations).

### 3.2. Description of Included Studies

Major characteristics of the included studies are shown in Table 1 and Table 2. Studies employed a qualitative design (*n* = 13), mixed-method design (*n* = 1), and quantitative design (*n* = 12). Quantitative studies included quantitative randomized studies (*n* = 4), meta-analysis study (*n* = 1), longitudinal quantitative study (*n* = 1), quantitative non-randomized study (*n* = 1), and others (*n* = 5). Studies were conducted across three of the seven continents, with most studies conducted in the Netherlands (*n* = 14, 53.8%) and the US (*n* = 4, 15.4%).

The included studies focused on child welfare (*n* = 10, 38.5%), mental health care (*n* = 8, 30.8%), youth justice (*n* = 2, 7.7%), disability care (*n* = 2, 7.7%), older adult healthcare (*n* = 2, 7.7%), homeless care (*n* = 1, 3.8%), and BME (Black and minority ethnic) immigrant background issues (*n* = 1, 3.8%).

Participants included the main actor and social network (families and community members), professionals, and coordinators. The qualitative studies comprised the main actor (*n* = 8, 61.5%), social-network members (*n* = 8, 61.5%), professionals (*n* = 10, 76.9%), and coordinators (*n* = 8, 61.5%). The quantitative and mixed-method studies comprised the patient, family members (*n* = 9, 69.2%), social-network members (*n* = 3, 23.1%), professionals (*n* = 9, 69.2%), and coordinators (*n* = 4, 30.8%).

In five papers, the effects of FGC over time were researched using the instrument, as shown in Table 1 and Table 2. Specifically, these studies assessed the effects of FGC after three months (*n* = 2), six months (*n* = 3), 12 months (*n* = 2), and 18 months (*n* = 1).

### 3.3. The Effectiveness of FGC

The effectiveness of FGC was extracted across 16 studies. The frequency of identification of each category of effectiveness is summarized in Table 3. The effectiveness of FGC included such categories as “sense of ownership” (*n* = 14), “restoring belongingness” (*n* = 10), “reduction of coercion” (*n* = 4), and “learning platform” (*n* = 4). One example of each subcategory is also shown in Table 3.

“Sense of ownership” refers to the idea of the main actor and their family making improvements and taking control of their situation through self-reflection during interactions with their social network in the FGC. This category includes the following subcategories: “self-reflection through FGC”, “users taking control of their situation”, and “improvement of users’ situations”.

“Restoring belongingness” refers to the idea of the main actor feeling safe and supported in their social network during the process of self-disclosure in FGC. This category includes the following subcategories: “self-disclosure during FGC process”, “feeling safe and accepted”, and “feeling supported”. “Sense of ownership” focuses on the initiative of the main actor and family to proactively reflect on their own situation and seek to improve it, while “restoring belongingness” refers to the emotional aspect of feeling a sense of security, acceptance, and support within a social network through self-disclosure in FGC, and restoring a sense of belonging.

“Reduction of coercion” refers to the process of the main actor’s self-respect and self-confidence improving through being connected or related to a social network, and also not feeling pressure exerted on them to accept treatment. This category includes the following subcategories: “improvement in self-respect” and “improvement in self-confidence”.

“Learning platform” refers to the partnership between the main actor and their social network, wherein the main actor can learn relational autonomy as a person from working in partnership with their social network. This category includes the following subcategories: “learning relational autonomy” and “learning to work with professionals”.

### 3.4. The Challenges to FGC Implementation

The challenges to FGC implementation were extracted across 18 studies. The frequency of identification of each category of the problems is summarized in Table 4. The challenges to FGC implementation included such categories as “severe situations of main actor” (*n* = 4), “severe situations of the family” (*n* = 6), “the complex role of the FGC coordinator” (*n* = 9), and “the cost-ineffectiveness of FGC” (*n* = 11). One example of each subcategory is shown in Table 4.

“Severe situations of main actor” refers to emergency situations where time is a critical factor, such as when a child requires immediate care, and situations in which the main actor is unable to make decision because of a serious illness. Both of these situations can pose difficulties in organizing and conducting FGC. This category includes the following subcategories: “emergency situations” and “severe problems of the main actor”.

“Severe situations of the family” refers to situations where relationships within family or community are broken or strained, such as poor inter-familial communication or dissolving network relationships. Such situations can reduce the likelihood of completing FGC. This category includes the following subcategories: “broken family” and “broken relationships within the network”.

“The complex role of the FGC coordinator” refers to the difficulties of the role of the FGC coordinator. FGC coordinators have to organize FGC and carry out FGC, and they need to consider the dynamics and unique situations of all of the participants. This category includes the following subcategories: “FGC coordinator complex role” and “difficulties encountered by coordinators”.

“The cost-ineffectiveness of FGC” refers to situations where FGC coordinators with experience or knowledge about the FGC process are insufficient, or where FGC spending is cost inefficient when compared to other models. This category includes the following subcategories: “lack of experience and knowledge of FGC implementation”, and “inefficient spending”.

## 4. Discussion

FGC differs from general decision-making models in that it places more emphasis on the interaction between participants and the reconstruction of social relationships than it places on the decision itself. While general models focus on the analysis and evaluation of clear options, in FGC, the process in which participants develop self-esteem and a sense of belonging and become independent decision-makers is considered important. Here, the ways in which interaction between the main actor and their social network can impact the implementation of FGC, both positively and negatively, will be discussed.

### 4.1. The Effectiveness of FGC and the FGC Process

FGC was effective in four aspects, but the literature works on “sense of ownership” and “restoring belongingness” were more numerous, suggesting these two aspects, in particular, may be more effective. The interaction between the main actor and social network in the FGC process appears to be a key factor in the effectiveness of FGC. According to Anderson and Parkinson (2018) [43], the interaction between these two key elements in FGC has a therapeutic effect, because they are able to discuss problems, issues, and difficulties. The relationship is not a one-way process that considers the perspective of the main actor alone but rather a two-way process that includes multiple perspectives from the social network as well, forming a reciprocal interaction [16]. In this way, the interactions between participants in FGC create the conditions for planning meetings [44], which enable the participants to solve their problems mutually, thereby forming the therapeutic foundation.

The interaction in FGC can improve the ability of participants to understand their situation, build relationships, and collaborate with others to change and improve their situation over time. Four categories of effects of FGC cannot be produced in isolation, but rather must be created through interpersonal social interaction. Therefore, the interaction of the two key elements does not facilitate decision-making in FGC per se but rather works on the participants to assist them in becoming decision-makers themselves.

### 4.2. Challenges to the Implementation of FGC and the FGC Process

The challenges facing the implementation of FGC can hinder the prospects for the FGC process to be successfully completed. In the course of research, it came to be understood that challenges to FGC implementation include both those that impede the successful completion of the FGC process itself and those that impede the very possibility of implementing FGC in the first place.

Regarding challenges that hinder the possibility of implementing FGC, De Jong et al. (2015) [31] analyzed the factors that block FGC preparations and the ability to plan FGC and implement it fully, such as sense of shame and sense of pride of the main actor, and a lack of initiative and care paralysis. The authors also noted that if the parties feel shame over their problems, this may lead to the fear that their social network will ascribe blame, ultimately hindering their participation in FGC. They also stated that, in these cases, the FGC preparation phase bogs down. These findings brought to light obstructions that may hinder the implementation of FGC, as well as the establishing of balanced relationships among participants.

On the other hand, according to Mitchell and Ambrose (2007) [45], numerous studies on reciprocal relationships focus on positive reciprocity, namely reciprocity that promotes stability in relationships through considerate, valued, and balanced exchanges. However, negative reciprocity can also affect relationships. For example, when a parent puts undue pressure on a child with depression, the child may react toward the parent with hostility, thus rendering meetings to resolve this problem unsuccessful. In this way, relationships with negative reciprocity can result in negative behavior. Negative reciprocity must be taken into account in the interaction between the main actor and their social network, and activities must also take into account the group dynamics of the participants.

This study focused mainly on the challenges faced during the FGC process. However, the challenges to effective FGC implementation involve not only FGC management that addresses the FGC process but also the time period leading up to the introduction of FGC. It is also necessary to identify the skills and qualifications of the coordinators who must take into account group dynamics, enabling the building of positive reciprocal interaction.

### 4.3. Strengths and Limitations

The first limitation of this review relates to the very small number of studies that give clear information about the time period of the FGC process and the method by which coordinators were selected. For this reason, the study’s findings may be frustrated by some shortcomings. The second limitation concerns the fact that more than half the authors were based in the Netherlands. However, this review included time-series analysis studies, and the Dutch authors’ studies were conducted in a variety of fields, including child abuse, community mental health, and disability. Therefore, despite the abovementioned limitations, the use of a scoping review method to provide a broad overview of FGC in numerous fields and in various forms, thereby enabling clarification of the effectiveness of FGC and the challenges in implementing FGC, works in this study’s favor. Future research should focus on the duration of FGC implementation and the method of selecting coordinators. In addition, in this study, the literature search was limited to the years 2015 to 2020, but ideally it would be desirable to review the literature from a wider range of search years. Moreover, the number of papers covered in this study was small (26), of which 14 were from the Netherlands. Taking these factors into consideration, the number of papers that can be discussed is limited, and it will be advisable to increase the number of papers referenced in the future.

### 4.4. Implications for Nursing Practice

It is difficult to implement FGC in all types of family nursing because many cases of decision-making support in the healthcare field require an emergency response. In this respect, it may be difficult to introduce FGC to all areas of nursing, particularly in acute or emergency care. However, it is believed that the effectiveness of the interaction between the main actor and social network, as revealed in this study, which focused on their abilities as decision-makers, can be practically utilized in nursing. Nurses have become involved in communication with healthcare providers, patients, and families in SDM. Inagaki (2020) [46] observed that this was limited to confirming the family’s understanding of information supplied by the healthcare provider and noted that the nursing professionals were hesitant to express their thoughts. In response to this, the following two recommendations are proposed:

Firstly, in FGC, the interaction between the main actor and the social network does not facilitate decision-making per se, but it works on the participants to assist them in becoming decision-makers themselves. Similarly, it is important to examine the effects of communication between healthcare providers, patients, and families in SDM on decision-making abilities, such as relational autonomy, rather than viewing it as simply a process of understanding information from healthcare providers, making choices, and agreeing. In addition to the participation by patients and their families, participation of those belonging to social networks, such as a friend or member of a kinship group, should also be considered.

Secondly, the coordinator in FGC plays an important role in shaping the interaction between the main actor and social network. In particular, the coordinator must plan and manage the FGC and take into account the dynamics of all participants. Similarly, in SDM, nursing professionals must take into account the dynamics of the participants and play the role of coordinator toward positive reciprocal relationships.

## 5. Conclusions

The effectiveness of FGC is exhibited by “sense of ownership”, “restoring belongingness”, “reduction of coercion”, and “learning platform”, which are created by the interaction between the main actor and social network, and this interaction impacts on the effectiveness of the capacity of decision-makers. The challenges to FGC implementation are “severe situations of main actor”, “severe situations of the family”, “the complex role of the FGC coordinator”, and “the cost-ineffectiveness of FGC”, all of which hinder the FGC process. However, the challenges to effective FGC implementation involve the FGC process and the time prior to the introduction of FGC, as well as the development of positive interactions between the main actors and their social networks. Because it has become clear that interactions between main actors and social networks are effective in decision-making in FGCs, we recommend that, after understanding the characteristics of FGCs, FGCs be utilized in clinical nursing to support decision-making for various families.

## Figures and Tables

**Figure 1 nursrep-15-00122-f001:**
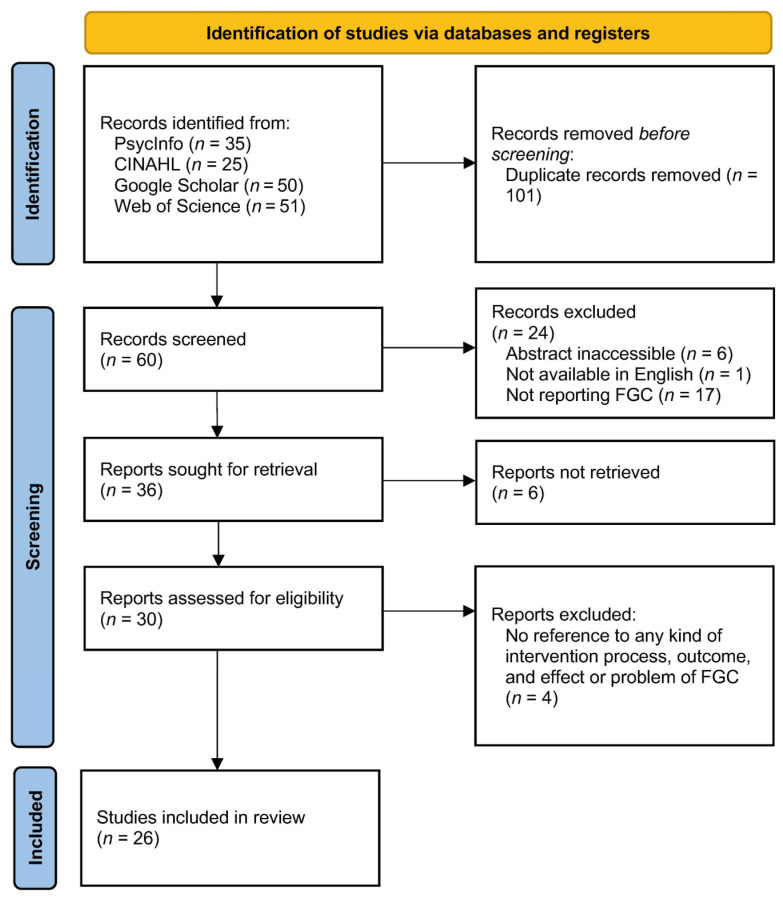
The flow of study identification and selection.

**Table 1 nursrep-15-00122-t001:** Characteristics of included qualitative publications (*n* = 13).

Reference Number, First Author’s Surname, Year of Publication, and Country	Participants				Study Design and Research Question	Key Findings
	MA	SN	P	C	Time-Series Analysis	
Child welfare						
[29] Mitchell (2020), Scotland	Children (*n* = 10), families (*n* = 22), and professionals (*n* = 28).				Semi-structured interviews and document analysis of FGC files were employed to retrospectively understand the contribution FGC makes to longer-term outcomes for children at risk of entering state care and their families.	FGC continuously shapes families’ capabilities and identities through emotional engagement, extending impact beyond meetings into daily life. Professionals assess outcomes by organizational priorities such as child-placement progress and meeting effectiveness.
	X	X	X	X	N/A	
[30] Schmid et al. (2017), Canada	FGC practitioners (*n* = 17).				Semi-structured, long telephone individual interviews and focus-group interviews were employed to identify the internal and external facilitative and inhibitory processes in promoting the shift to FGC use, opportunities, and threats to the program, as well as the processes that would lead to continued sustainability of the program.	FGC program’s success stems from multifaceted reforms, yet long-term sustainability faces challenges due to unstable funding and uneven support across provincial and agency levels.
		X	X	X	N/A	
**Mental health**						
[8] Schout et al. (2017), The Netherlands	Eighteen cases of FGC.				Semi-structured interviews were completed using a responsive evaluation methodology to consider the possibility of collecting feedback as a way to contribute positively to the alliance between FGC coordinators and those for whom FGC is deployed.	Families persisted in avoiding care pre-/post-FGC, while legal guardians disengaged and coordinators’ inertia stalled interventions. Feedback-driven strategies may mitigate emerging care avoidance and paralysis.
	X	X	X	X	N/A	
[9] Schout et al. (2017), The Netherlands	Seventeen cases with psychiatric problems, which were presented by psychiatrists (*n* = 4) and community mental health nurses (*n* = 2).				Interviews were used to elucidate in what circumstances FGC cannot be deployed.	Key barriers: Time pressure; the severity of the mental state of clients; professionals’ difficulties to consider, or inability to visualize, FGC; and lack of receptivity by clients and/or networks to FGC.
	X		X		N/A	
[10] De Jong et al. (2018), The Netherlands	Forty-one cases of FGC				Semi-structured interviews were conducted to examine the process and impact of the conferences.	FGC dynamics: Resistance/isolation resolution, shared emotion disclosure, maternal-driven motivation (vs. professionals), coordinator role complexity, and professionals’ non-interference. Key factors: To invite people and extend their social network; to share shameful feelings and grievances; trust between clients and FGC coordinators; and professionals’ reinforcement of the self-direction of FGC.
	X	X	X	X	N/A	
[12] Johansen (2020), Norway	Nine men and six women.				Semi-structured interviews were conducted to explore long-term social-assistance recipients’ experiences with FGC.	Three core therapeutic network mechanisms emerged: self-disclosure, confronting and improving unsatisfactory family relations, and dialogic communication.
	X				N/A	
[31] De Jong et al. (2015), The Netherlands	Main actors (clients) (*n* = 29), people from the social network (family, friends, and neighbors) (*n* = 35), professionals (social workers, mental health nurses, police officers, employees of housing associations, and municipalities) (*n* = 37), and FGC coordinators (*n* = 17).				Semi-structured interviews were conducted to examine the process and impact of the conferences.	FGC was predominantly deployed post-professional-care failure, yet it fell short of its objectives due to underutilized social networks and clients’ perceived helplessness in effecting change.
	X	X	X	X	N/A	
[32] Meijer et al. (2019), The Netherlands	Attendees of FGC (*n* = 289).				Interviews and participant observations were conducted to understand the process and impact of the FGC.	Coercive psychiatry FGC outcomes: Ownership (the feeling of having control) over the situation and taking the initiative after the FGC and expanded support networks. Partnership conditions: Vulnerability/shame disclosure, conflict avoidance, mental health professionals’ attitudinal adaptation, and the facilitating role of the coordinator.
	X	X	X	X	Interviews were conducted one-to-six months after FGC.	
**Care for older adults**						
[15] Górska et al. (2016), Scotland	Families (*n* = 14) and professionals participating in FGC.				Focus-group interviews were conducted to evaluate the impact of the pilot FGC service in dementia care.	Benefits: Enables families to collaboratively address dementia care needs through synchronized, purpose-driven gatherings. Challenges: Dementia-related cognitive impairments may limit individuals’ capacity for informed participation consent.
		X	X		N/A	
[16] Metze et al. (2015), The Netherlands	Older adults (*n* = 8), social network members (*n* = 4), and social workers (*n* = 4), and two contrasting FGC cases.				Case-study design was employed to explore the appropriateness of FGC in older adults in terms of resilience and relational autonomy.	The concepts of relational autonomy and resilience provide insight into the FGC process. Compassionate interventions and respect for elders’ needs empower proactive problem-solving.
	X	X	X		N/A	
**Youth justice**						
[33] Slater et al. (2015), New Zealand	Youth justice coordinators (*n* = 19), and practitioners with a range of experience (*n* = 27).				Semi-structured interviews were conducted to understand the development of practice and to identify factors constituting best practice and areas of process weakness.	Youth justice FGC effectively reduced recidivism in most cases but underperformed for high-risk re-offenders. Best practice included aligning professional approaches to FGC philosophy and practice, training coordinator-led delivery, FGC preparation quality, victim inclusion, and strengths-based personalized plans.
				X	N/A	
**Black and minority ethnic (BME) families**						
[34] Valenti (2017), Scotland	Professionals (*n* = 8), of whom 6 considered themselves to be from a BME background.				A review of the literature and a series of interviews were used to explore the use of FGC in social work with children and families from BME backgrounds.	FGC remains under-researched and underutilized with BME families. Mandatory referrals could enhance BME families’ participation in decision-making, yet practitioners report systemic challenges, with interpreter reliance and complex family dynamics requiring solutions.
			X	X	N/A	
**Homelessness problem**						
[13] Miklosko et al. (2017), Slovakia	Clients with homelessness problems (*n* = 42) and professionals (*n* = 16).				Semi-structured interviews were conducted to discover the impact of FGC on families with a homelessness problem.	FGC-driven factors that reinforced families were understanding familial problem contextualization; reduced social isolation, supporting the establishment of new relations for the family; renewed relational bonds; expanded support networks; improved relations in the family system; and six other factors.
	X	X	X		N/A	

Note: MA = main actor; SN = social network; P = professional; C = coordinator; N/A = not applicable; FGC = Family Group Conferencing.

**Table 2 nursrep-15-00122-t002:** Characteristics of included quantitative and mixed-method publications (*n* = 13).

Reference, First Author’s Surname, Year of Publication, and Country	Participants				Study Design and Outcome Measures	Key Findings
	MA	SN	P	C	Time-Series Analysis	
Child welfare						
[5] Sen et al. (2019), UK	National data and quantitative data from a study of FGC service in a city.				Quantitative research was employed to measure the changes in a looked-after child (LAC), a child protection plan (CP), and a child in need (CIN) rates in FGC service in a city compared with those of the nation overall.	The city’s LAC rate, initially above the national average, has declined since 2012, coinciding with restorative practice adoption. CP rates showed steady decline post-2013, falling below the national average by 2015. While CIN rates exhibited volatility, they consistently exceeded the national average from 2011 to 2016.
					N/A	
[24] Dijkstra et al. (2016), The Netherlands	Fourteen controlled studies (*n* = 88,495 participants).				A meta-analytic study was conducted to examine the effects of FGC on child safety (in terms of reports of child maltreatment and out-of-home placement) and involvement of youth care.	FGC did not significantly reduce child maltreatment, out-of-home placements, or youth care involvement. Retrospective studies found it more effective than standard care in reducing recurrence and placement duration, whereas prospective studies showed weaker efficacy.
					N/A	
[35] Corwin et al. (2020), USA	Families (*n* = 287) with a substantiated report of child abuse or neglect assigned to receive in-home services. Caseworkers completed a case-specific questionnaire (CSQ), which contained questions related to service needs and service provision for families, improvements experienced by families, and other case specifics.				A randomized controlled trial (RCT) was conducted. Dependent variable was the perceived improvement in social support contained in CSQ. Independent variables were whether or not a family participated in FGC, whether a family was assigned to the treatment or control group, the race/ethnicity, age, person type, and so on.	Families participating in FGC had 4.46 times higher odds of improved social support than controls. Each additional child increased caseworkers’ likelihood of reporting improved support by 17%, suggesting FGC enhances social networks and potentially safeguards child welfare.
			X		N/A	
[36] Dijkstra et al. (2017), The Netherlands	Families (*n* = 229) with problems in different domains, such as delinquency, school problems, child maltreatment, mental health, alcohol and drug problems, and high-conflict divorce.				Univariate logistic regression analyses were performed to examine whether demographic characteristics, parent characteristics, and family characteristics affected the willingness to organize FGC and the likelihood of actually accomplishing FGC.	While 60% of families initially agreed to FGC participation, only 27% completed it. Attrition stemmed from motivational deficits, high-conflict divorces, or competing care priorities, with fragmented or newly formed families demonstrating markedly lower completion rates.
	X		X	X	N/A	
[37] Dijkstra et al. (2018), The Netherlands	Experimental group (*n* = 46 families), and control group (*n* = 23 families).				RCT was conducted. Outcome measures were Actuarial Risk Assessment Instrument Youth Protection (ARIJ) to assess child maltreatment; Family Empowerment Scale (FES); short version of the Interpersonal Support Evaluation List (ISEL-short form); a cost questionnaire; and unit costs.	FGC proved cost-ineffective for child safety, empowerment, and social support—with effectiveness varying by completion levels—with negligible cost differences versus standard care.
					Data were collected at pretest, and one, three, six and 12 months after a care plan had been made.	
	X		X			
[38] Dijkstra et al. (2019), The Netherlands	Experimental group (*n* = 229 families) and care as usual group (*n* = 99 families).				RCT was conducted. Outcome measures were child safety score; risk of child maltreatment by child welfare worker and the parents; out-of-home placement and supervision order extracted from case file reports; the number of professional services used; Family Empowerment Scale (FES).	While FGC matched usual care in improving child safety, it led to more out-of-home placements, prolonged child welfare involvement, and slightly higher service utilization, as opposed to enhanced parental empowerment and social support.
					Data were collected at pretest and one, three, six, and twelve months after a care plan had been made.	
[39] Hollinshead et al. (2017), USA	Treat group (*n* = 248 families) and control group (*n* = 255 families).				RCT was conducted. Outcome measures were re-referrals to child protective services, substantiated re-referrals, and out-of-home placements.	FGC participation showed no significant impact on re-referral, substantiated re-referral, or out-of-home placement odds; however, families with multiple children or parents faced elevated re-referral and a substantiated re-referral risk.
	X		X		N/A	
[40] Merkel-Holguin et al. (2020), USA	Ten children/youth of interest, 678 family/fictive kin, and 121 professionals.				Quantitative research was employed using fidelity index, which consists of three subscale scores of family leadership, inclusion and respect preparedness, and transparent planning.	Families/fictive kin perceived lower fidelity achievement across domains than professionals, with children/youth expressing the lowest agreement.
	X	X	X		N/A	
**Disability healthcare**						
[14] Brongers et al. (2020), The Netherlands	Nine clients participated in FGC.				A mixed-method pre- post-intervention feasibility study was employed using questionnaires, semi-structured interviews, and return-to-work plans drafted in FGC. Feasibility outcomes were demand, acceptability, implementation, and limited efficacy of perceived mental health and level of participation.	FGC participants reported high satisfaction, with slight improvements in mental health and participation during follow-up. Most return-to-work-plan actions focused on employment goals. Client-led, socially supported employment actions (post-FGC) enabled 5 participants to re-enter paid/voluntary work within 6 months.
					Data were collected directly after and then three and six months after FGC.	
	X		X	X		
[41] Onrust et al. (2015), The Netherlands	Anonymized file data collected from 71 clients who had taken part in FGC and a comparable group of 53 clients who had not.				Quantitative study was employed to measure child functioning, family/child-rearing environment, and wider environment.	FGC participants showed sharper problem reduction over 12 months vs. non-participants’ moderate decline, with comparable resource-use between groups.
					The areas of concern were assessed before and about 12 months after FGC.	
	X					
**Public mental health care**						
[7] De Jong et al. (2016), The Netherlands	Main actor (*n* = 74), social network (*n* = 119), professionals (*n* = 77), and FGC coordinator (*n* = 42).				Quantitative study was employed to measure social support, resilience, and living conditions.	FGC implementation enhanced social support, resilience, and living conditions for clients with baseline resource scarcity and constrained networks.
					Data were collected within one to six months after FGC.	
	X	X	X	X		
[11] Meijer et al. (2017), The Netherlands	Client (*n* = 33), social network (*n* = 135), professionals (*n* = 56), and FGC coordinator (*n* = 29).				A responsive evaluation, including qualitative and quantitative methods, was employed to measure belongingness, ownership, and coercion.	Belongingness/ownership demonstrated significant post-FGC growth, contrasting with marginally reduced coercion.
					Data were collected between 7 and 18 months after FGC.	
	X	X	X	X		
**Youthful offenders**						
[42] Hipple et al. (2015), USA	FGC (*n* = 215) described in two data sources from the Indianapolis Juvenile Restorative Justice Experiment.				Quantitative study was employed to measure failure and elements of restorativeness.	Restorative conferences reduced long-term failure rates from 99% to 71% for violent offenders and from 98% to 54% for non-violent offenders, compared to non-restorative approaches.
					N/A	

Note: MA = main actor; SN = social network; P = professional; C = coordinator; N/A = not applicable; FGC = Family Group Conferencing.

**Table 3 nursrep-15-00122-t003:** Summary of findings on effectiveness of Family Group Conferencing.

Identified Categories	Identified Subcategories	One Example and Its Reference Number	References
Sense of ownership	Self-reflection through FGC	“With the help of his social network, he [the main actor] is able to look at his actions and to realize that they limit his wellbeing. He knows he needs a push to come into action and improve his life. He can deal with criticism, if it is constructive and given from the heart” [16].	[7,10,11,13,14,15,16,29,30,32,33,34,38,42]
	Users taking control of their situation	“Several clients mentioned that the FGC contributed to their feelings of ownership (the feeling of having control) over the situation. For example, appointments and agreements with family and friends gave more structure to the life of clients, served as an extra motivation …” [11].	
	Improvement of users’ situations	“Children and family members interviewed in the study considered their outcomes were linked to their personal experience of FGC. Families expressed outcomes in terms of process and/or learning and/or a change in their quality of life” [29].	
Restoring belongingness	Self-disclosure during FGC process	“… the informants chose to disclose personal information at the FGC. Sometimes this information was more or less unknown to their family and friends. Some told honestly about how their everyday life in fact …” [12].	[7,10,11,12,13,29,32,33,35,38]
	Feeling safe and accepted	“A sense of acceptance by the wide family and wider social network” [13].	
	Feeling supported	“This [belongingness] was measured with a scale question about the perceived social support before and after the conference [FGC]… This outcome is also significant and thus demonstrates that the respondents were of the opinion that the ‘sense of belongingness’ had increased after the conferences” [11].	
Reduction of coercion	Improvement in self-respect	“Before, during and after the FGC, he is surrounded by people who appreciate his openness, who notice and mention his positive changes, and who worry about and think along with him” [16].	[11,13,16,29]
	Improvement in self-confidence	“Respondent’s interviews resonated with how the process had made people feel respected, supported, valued, and acknowledged (or not) by their extended family, during and because of the FGC experience” [29].	
Learning platform	Learning relational autonomy	“The presence of his friends and family during the FGC gives him more self-esteem. He feels loved and this helps him to take initiatives again and to invite people to undertake some collective activity, thus making his relationships more reciprocal” [16].	[13,15,16,29]
	Learning to work with professionals	“The FGC was perceived as a platform for sharing the family story with professionals and presenting them with a comprehensive view of the service user’s personal circumstances, therefore facilitating development of a better understanding of the individual and his/her needs” [15].	

**Table 4 nursrep-15-00122-t004:** Summary of findings on challenges to Family Group Conferencing implementation.

Identified Categories	Identified Subcategories	One Example and Its Reference Number	References
Severe situations of main actor	Emergency situations	“In crisis situations there is little time to organize an FGC” [9].	[9,15,33,42]
	Severe problems of the main actor	“It was seen as particularly challenging when a person with dementia had limited insight into his/her difficulties and when more sensitive issues regarding the person’s needs were discussed” [15].	
Severe situations of the family	Broken family	“Poor inter-familial communication patterns were said to often result in a propensity for anger and/or violence, and often aligned with drug and alcohol abuse. A lack of basic literacy and numeracy skills was a key factor believed to influence the typically low self-esteem” [33].	[8,31,33,34,36,41]
	Broken relationships within the network	“Often contacts between clients and their network were so heavily damaged or had faded, becoming so attenuated that family and bystanders were reluctant to participate in a conference” [31].	
The complex role of the FGC coordinator	FGC coordinator complex role	“Study participants described an effective Co-ordinator skill-set as including facilitation, mediation, conflict resolution, negotiating abilities, motivational interviewing techniques, and navigating group dynamics (including handling strong emotions)” [33].	[8,9,10,15,32,33,36,38,40]
	Difficulties encountered by coordinators	“FGC coordinators frequently struggled with their role as facilitator (who aim to give clients the power to determine their own life), especially when clients remained passive and the social network was kept away” [10].	
The cost-ineffectiveness of FGC	Lack of experience and knowledge of FGC implementation	“FGC is not yet in routine. No sufficient knowledge and experience” [9].	[5,9,24,30,31,33,36,37,38,39,41]
	Inefficient spending	“The use of FGC led to larger effects, but also to higher costs, while the chance that FGC would be more cost-effective than CAU was 30% (33% with an investment of 10.000 euro)” [36].	

## Data Availability

The data presented in this study are available upon request from the corresponding author.

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
