# Peer review of "The Effectiveness of Family Group Conferencing and the Challenges to Its Implementation: A Scoping Review"

_nursrep, 2025, doi:10.3390/nursrep15040122_

Round 1
Reviewer 1 Report
Comments and Suggestions for Authors
The manuscript is based on the effectiveness of Family Group Conferencing (FGC) and the challenges associated with its implementation through a scoping review. The topic is very relevant to family nursing and social care, and the study provides valuable insights into FGC’s strengths and limitations. However, I think that yoy must increase areas of improvement to enhance the clarity, organization, and scientific rigor of the paper.
Introduction
The introduction is well written but you could insert a more structured discussion of the research gap and the significance of the study. You can explain the differences between FGC and other decision-making frameworks to strengthen the rationale of the study.
Methods
The method section is well-written, but you must add details are on the search strategy, selection criteria, and data extraction process.
You should justify the chosen of time frame (2015–2020) and explain any potential limitations associated with it.
Please, clarify whether a risk of bias assessment was conducted for the included studies.
Results
The presentation of results is well-structured, but you must refine some tables and figures for better readability. Some categories in the thematic analysis are broad; providing more specific examples from the studies reviewed would improve clarity.
Discussion
The discussion could be more focused and concise. Some parts repeat information already presented in the results. A deeper comparison with existing literature on decision-making models in family and community health care would strengthen the argument.
Consider addressing potential biases and methodological limitations more explicitly.
Conclusion
The conclusions align with the findings, but you must insert more impactful of practical implications for policymakers, practitioners, and researchers.
Comments on the Quality of English Language
The manuscript contains some unclear or awkwardly worded sentences. A professional English revision would improve clarity and readability.
Author Response
Comments 1: The manuscript is based on the effectiveness of Family Group Conferencing (FGC) and the challenges associated with its implementation through a scoping review. The topic is very relevant to family nursing and social care, and the study provides valuable insights into FGC’s strengths and limitations. However, I think that yoy must increase areas of improvement to enhance the clarity, organization, and scientific rigor of the paper.
Introduction
The introduction is well written but you could insert a more structured discussion of the research gap and the significance of the study. You can explain the differences between FGC and other decision-making frameworks to strengthen the rationale of the study.
Response 1: In lines 108 to 120, we have already explained the research gap and the significance of the study. Moreover, the following was added to line 120:
As research progresses into FGC, new methods of family decision-making may be developed.
In lines 92 to 107 moreover, we have already explained the differences with shared decision making (SDM). As pointed out, the following has been added to line 39: This is different from the general idea of decision making.
In addition, the following was added to line 68: Having professionals involved with families at this stage is different from the general method of decision making.
Comments 2: Methods
The method section is well-written, but you must add details are on the search strategy, selection criteria, and data extraction process.
You should justify the chosen of time frame (2015–2020) and explain any potential limitations associated with it.
Please, clarify whether a risk of bias assessment was conducted for the included studies.
Response 2: The search strategy has been described in more detail in the text. From line 174 onwards, the following corrections have been made:
The search identified 161 articles: PsycInfo (n = 35); CINAHL (n = 25), Google Scholar (n = 50), and Web of Science (n = 51). After removing 101 duplicate articles, 60 abstracts were screened, and then those papers which were abstract inaccessible (n = 6); not available in English (n = 1); and not reporting FGC (n = 17) were excluded. Thirty full papers were accessed for further evaluation based on the exclusion criteria (no reference to any kind of intervention process, outcome, and effect or problem of FGC), resulting in the final review of 26 full-text articles (43.3% of identified citations).
The reason for limiting the search years to 2015-2020 is stated on lines 144 to 146 and 53 to 55. In addition, the following has been added from line 338 as a limitation of the study.
In addition, in this study, the literature search was limited to the years 2015 to 2020, but ideally it would be desirable to review literature from a wider range of search years.
Comments 3: Results
The presentation of results is well-structured, but you must refine some tables and figures for better readability. Some categories in the thematic analysis are broad; providing more specific examples from the studies reviewed would improve clarity.
Response 3: Taking into consideration the comment by the other reviewer, because tables 1 and 2 were too long, we summarized and shortened the "Key findings."
Comments 4: Discussion
The discussion could be more focused and concise. Some parts repeat information already presented in the results. A deeper comparison with existing literature on decision-making models in family and community health care would strengthen the argument.
Consider addressing potential biases and methodological limitations more explicitly.
Response 4: As was pointed out, the section summarizing the results was deleted.
Regarding the differences from general decision-making models, we have added the following from line 269:
FGC differs from general decision-making models in that it places more emphasis on the interaction between participants and the reconstruction of social relationships than it places on the decision itself. While general models focus on the analysis and evaluation of clear options, in FGC, the process in which participants develop self-esteem and a sense of belonging and become independent decision-makers is considered important.
Regarding potential biases and methodological limitations, these were mentioned from line 326 to line 338. This is the same as Response 2, but with the following added to line 338:
In addition, in this study, the literature search was limited to the years 2015 to 2020, but ideally it would be desirable to review literature from a wider range of search years. Moreover, the number of papers covered in this study was small (26), of which 14 were from the Netherlands. Taking these factors into consideration, the number of papers that can be discussed is limited, and it will be advisable to increase the number of papers referenced in the future.
Comments 5: Conclusion
The conclusions align with the findings, but you must insert more impactful of practical implications for policymakers, practitioners, and researchers.
Response 5: From line 377, the following has been added:
Because it has become clear that interactions between main actors and social networks are effective in decision-making in FGCs, we recommend that, after understanding the characteristics of FGCs, FGCs be utilized in clinical nursing to support decision-making for various families.
Reviewer 2 Report
Comments and Suggestions for Authors
Thank you so much for the opportunity to read your paper. I have few comments you might consider or clarify:
The manuscript presents effectiveness dimensions (e.g., “Sense of ownership,” “Restoring belongingness,” “Reduction of coercion,” “Learning platform”) and challenges (e.g., “Severe situations of main actor,” “The complex role of the FGC coordinator,” “Cost‑ineffectiveness”). Although these categories are clearly extracted from the included studies, it is not always clear whether the categorization was driven solely by frequency counts or if a deeper thematic synthesis was attempted. For example, the differentiation between “Sense of ownership” and “Restoring belongingness” could be further explained. Clarifying the conceptual boundaries between these categories would strengthen the interpretation.
The authors report frequencies for each category; however, given the limited number of studies (26) and the geographic concentration (with more than half from the Netherlands), caution should be used when generalizing the effectiveness of FGC. There is a risk of overestimating positive outcomes if the sample is not representative of broader international practice. The discussion should include a critical reflection on this potential bias and how it might influence conclusions.
The review includes studies using diverse methodologies (qualitative, quantitative, and mixed methods), yet the synthesis sometimes reads as a listing of findings rather than an integrated analysis. Although tables (Tables 1–4) help summarize individual study characteristics and extracted themes, the text could more clearly explain how the two strands of evidence complement each other to inform the overall conclusions regarding FGC effectiveness and challenges.
The tables are comprehensive but, at times, overly dense. For instance, Table 1 and Table 2 list extensive details about study characteristics, yet some columns (e.g., “Key findings”) might benefit from a more concise presentation. It is important to ensure that the reader can quickly grasp the key points without being overwhelmed by information.
There are minor inconsistencies in formatting (e.g., abbreviations like “FGC” are used appropriately, but a few abbreviations or symbols are not always defined on first use). Also, ensure that percentages, numbers, and labels are consistent throughout. For example, check that sample sizes and subgroup percentages are uniformly reported.
Flow Diagram (Figure 1):
The PRISMA flow diagram is a critical component of a scoping review. Verify that it clearly maps the search and selection process and that its labeling is consistent with the text. Any supplementary figures mentioned in the text (such as those detailing time series analyses) should be clearly referenced and easy to interpret.
While you follow established frameworks (Arksey and O’Malley, PRISMA-ScR), more detail on how themes were inductively derived would be useful. For example, a brief description of the coding process and inter-rater reliability (if applicable) would help readers assess the rigor of the synthesis.
The criteria for study inclusion (e.g., why only articles with “family group conferencing” in the title were chosen) may introduce bias. Addressing potential limitations stemming from these search parameters would be beneficial.
Ensure consistent use of key terms. For example, if “main actor” and “social network” are central to the discussion, consider including a clear, early definition in the methods or introduction.
Some sections, especially in the discussion, repeat similar points regarding the interaction between the main actor and their social network. Consolidating these repetitive elements could improve readability and strengthen your argument.
Reference and Citation Issues:
In a few places, the text mentions author names with dates (e.g., “[16]”) without a clear connection to the reference list. Ensure that all in-text citations are correctly formatted
Metze et al. (2015b) ? a,b repeated references?
Best wishes
Author Response
Comments 1: Thank you so much for the opportunity to read your paper. I have few comments you might consider or clarify:
The manuscript presents effectiveness dimensions (e.g., “Sense of ownership,” “Restoring belongingness,” “Reduction of coercion,” “Learning platform”) and challenges (e.g., “Severe situations of main actor,” “The complex role of the FGC coordinator,” “Cost‑ineffectiveness”). Although these categories are clearly extracted from the included studies, it is not always clear whether the categorization was driven solely by frequency counts or if a deeper thematic synthesis was attempted. For example, the differentiation between “Sense of ownership” and “Restoring belongingness” could be further explained. Clarifying the conceptual boundaries between these categories would strengthen the interpretation.
Response 1: The analytical method is described in lines 151 to 171. This time, the following has been added:
Descriptive data from individual studies were collated, and the effectiveness and challenges to FGC were identified for each context. Next, each description was divided into small, meaning units, or codes. Code classification was performed by focusing on similarities and differences, with codes found to be conceptually similar grouped into more abstract concepts termed subcategories. The subcategories were then grouped to create categories and their abstraction levels were refined and checked by the review team.
The difference between "Sense of ownership" and "Restoring belongingness" is described in lines 217 to 225. Furthermore, the following comment has been added on line 225:
"Sense of ownership" focuses on the initiative of the main actor and family to proactively reflect on their own situation and seek to improve it, while "Restoring belongingness" refers to the emotional aspect of feeling a sense of security, acceptance, and support within a social network through self-disclosure in FGC, and restoring a sense of belonging.
Comments 2: The authors report frequencies for each category; however, given the limited number of studies (26) and the geographic concentration (with more than half from the Netherlands), caution should be used when generalizing the effectiveness of FGC. There is a risk of overestimating positive outcomes if the sample is not representative of broader international practice. The discussion should include a critical reflection on this potential bias and how it might influence conclusions.
Response 2: From line 340, as a limitation of this study the following has been added:
Moreover, the number of papers covered in this study was small (26), of which 14 were from the Netherlands. Taking these factors into consideration, the number of papers that can be discussed is limited, and it will be advisable to increase the number of papers referenced in the future.
Comments 3: The review includes studies using diverse methodologies (qualitative, quantitative, and mixed methods), yet the synthesis sometimes reads as a listing of findings rather than an integrated analysis. Although tables (Tables 1–4) help summarize individual study characteristics and extracted themes, the text could more clearly explain how the two strands of evidence complement each other to inform the overall conclusions regarding FGC effectiveness and challenges.
Response 3: Tables 1 and 2 introduce the literature that was analyzed. Tables 3 and 4 show the results of the analysis, With the analysis method described in lines 151 to 171. From line 161, the following has been added:
Descriptive data from individual studies were collated, and the effectiveness and challenges to FGC were identified for each context. Next, each description was divided into small, meaning units, or codes. Code classification was performed by focusing on similarities and differences, with codes found to be conceptually similar grouped into more abstract concepts termed subcategories. The subcategories were then grouped to create categories and their abstraction levels were refined and checked by the review team.
Comments 4: The tables are comprehensive but, at times, overly dense. For instance, Table 1 and Table 2 list extensive details about study characteristics, yet some columns (e.g., “Key findings”) might benefit from a more concise presentation. It is important to ensure that the reader can quickly grasp the key points without being overwhelmed by information.
Response 4: In Tables 1 and 2, "Key findings" were abbreviated.
Comments 5: There are minor inconsistencies in formatting (e.g., abbreviations like “FGC” are used appropriately, but a few abbreviations or symbols are not always defined on first use). Also, ensure that percentages, numbers, and labels are consistent throughout. For example, check that sample sizes and subgroup percentages are uniformly reported.
Response 5: Every effort was made to confirm the contents thoroughly, but no revisions are warranted.
Comments 6: Flow Diagram (Figure 1):
The PRISMA flow diagram is a critical component of a scoping review. Verify that it clearly maps the search and selection process and that its labeling is consistent with the text. Any supplementary figures mentioned in the text (such as those detailing time series analyses) should be clearly referenced and easy to interpret.
Response 6: We confirmed that there are no problems with the PRISMA flow diagram. Also, the search strategy has been described in more detail in the text. From line 174 onwards, the following corrections have been made:
The search identified 161 articles: PsycInfo (n = 35); CINAHL (n = 25), Google Scholar (n = 50), and Web of Science (n = 51). After removing 101 duplicate articles, 60 abstracts were screened, and then those papers which were abstract inaccessible (n = 6); not available in English (n = 1); and not reporting FGC (n = 17) were excluded. Thirty full papers were accessed for further evaluation based on the exclusion criteria (no reference to any kind of intervention process, outcome, and effect or problem of FGC), resulting in the final review of 26 full-text articles (43.3% of identified citations).
The presence or absence of time series analysis and relevant details are indicated as literature information in Tables 1 and 2.
Comments 7: While you follow established frameworks (Arksey and O’Malley, PRISMA-ScR), more detail on how themes were inductively derived would be useful. For example, a brief description of the coding process and inter-rater reliability (if applicable) would help readers assess the rigor of the synthesis.
The criteria for study inclusion (e.g., why only articles with “family group conferencing” in the title were chosen) may introduce bias. Addressing potential limitations stemming from these search parameters would be beneficial.
Response 7: Details about the coding process etc. were already described on lines 161 to 171.
"Family group conferencing" as appearing the title was a mistake on our part. On lines 143 and 144, The sentence "Criteria were set at the outset of the study and consistent terms were 'family group conferencing' or 'family group conference' in the title" has been corrected to "The search keywords were set to 'family group conferencing' or 'family group conference.'"
Comments 8: Ensure consistent use of key terms. For example, if “main actor” and “social network” are central to the discussion, consider including a clear, early definition in the methods or introduction.
Some sections, especially in the discussion, repeat similar points regarding the interaction between the main actor and their social network. Consolidating these repetitive elements could improve readability and strengthen your argument.
Response 8: The definitions of "main actor" and "social network" are provided on lines 79 to 81.
As "main actor" and "social network" are important terms in FGC, these terms are used repeatedly. As can be seen, however, their number has been reduced throughout the discussion.
Comments 9: Reference and Citation Issues:
In a few places, the text mentions author names with dates (e.g., “[16]”) without a clear connection to the reference list. Ensure that all in-text citations are correctly formatted
Metze et al. (2015b) ? a,b repeated references?
Best wishes
Response 9: "Metze et al. (2015b) [16]" on line 94 was changed to "Metze et al. (2015) [16]." "De Jong et al. (2015b) [31]" on line 289 was
changed to "De Jong et al. (2015) [31]."
Round 2
Reviewer 1 Report
Comments and Suggestions for Authors
The manuscript presents a comprehensive and timely scoping review on the effectiveness and challenges of Family Group Conferencing (FGC). It provides a useful categorization of outcomes and obstacles, which can support both academic and clinical audiences.
Strengths:
• Clear adherence to scoping review methodology.
• Well-developed tables and thematic categorizations.
• Relevant discussion connecting FGC to broader healthcare and decision-making models.
Areas for Improvement:
1. Abstract: Consider simplifying sentence structure and clearly stating the number of studies included in the final analysis.
2. Introduction: Divide long paragraphs into smaller units to enhance readability. Strengthen the rationale by connecting FGC to clinical nursing more explicitly.
3. Discussion: Some sections are repetitive; streamline key ideas on interaction and relational autonomy to avoid redundancy.
4. Language: A few grammatical refinements and stylistic improvements are suggested, especially in phrasing such as “This review was comprised of…” (should be “included”) and “11 third researchers” (consider rewording).
5. Implications: The nursing practice implications are appreciated; consider further elaboration on how nurses might apply FGC principles in acute or emergency care contexts.
Comments on the Quality of English Language
A few grammatical refinements and stylistic improvements are suggested, especially in phrasing such as “This review was comprised of…” (should be “included”) and “11 third researchers” (consider rewording).
Author Response
Comments 1: The manuscript presents a comprehensive and timely scoping review on the effectiveness and challenges of Family Group Conferencing (FGC). It provides a useful categorization of outcomes and obstacles, which can support both academic and clinical audiences.
Strengths:
• Clear adherence to scoping review methodology.
• Well-developed tables and thematic categorizations.
• Relevant discussion connecting FGC to broader healthcare and decision-making models.
Areas for Improvement:
1. Abstract: Consider simplifying sentence structure and clearly stating the number of studies included in the final analysis.
Response 1: We have made revisions to the Abstract as per the recommendations. As for the number of studies included in the final analysis, the number was 26, as appears on line 19.
Comments 2: 2. Introduction: Divide long paragraphs into smaller units to enhance readability. Strengthen the rationale by connecting FGC to clinical nursing more explicitly.
Response 2: With regard to the Introduction, we have made revisions as per the recommendations wherever possible. As concerns rationale by connecting FGC to clinical nursing, this was already mentioned, but on line 108 we have added the text below:
Specifically, in clinical nursing practice, FGC provides nurses with an effective framework to foster patient and family engagement, enhance communication among care teams, and support holistic, patient-centered care through empowerment and relational autonomy. Moreover, nurses, who typically work closely with patients and their families, are ideally positioned to facilitate FGC, thereby improving care coordination, adherence to treatment plans, and overall health outcomes by leveraging families' inherent strengths and social networks.
Comments 3: 3. Discussion: Some sections are repetitive; streamline key ideas on interaction and relational autonomy to avoid redundancy.
Response 3: With regard to Discussion, all possible efforts were made to follow the suggestion.
Comments 4: 4. Language: A few grammatical refinements and stylistic improvements are suggested, especially in phrasing such as “This review was comprised of…” (should be “included”) and “11 third researchers” (consider rewording).
Response 4: "This review was comprised of..." on line 19 has been changed to "This review included..."
The "11 third researchers" on line 178 is a reference to researchers other than the authors. For clarity we have changed it to "11 collaborating researchers."
Comments 5: 5. Implications: The nursing practice implications are appreciated; consider further elaboration on how nurses might apply FGC principles in acute or emergency care contexts.
Response 5: An explanation for this already appears from line 348. To this, we have added: "In this respect, it may be difficult to introduce FGC in all areas of nursing, particularly in acute or emergency care."
Comments 6: Comments on the Quality of English Language
A few grammatical refinements and stylistic improvements are suggested, especially in phrasing such as “This review was comprised of…” (should be “included”) and “11 third researchers” (consider rewording).
Response 6: "This review was comprised of..." on line 19 has been changed to "This review included..."
The "11 third researchers" on line 178 is a reference to researchers other than the authors. For clarity we have changed it to "11 collaborating researchers."
Reviewer 2 Report
Comments and Suggestions for Authors
Thank you for addressing the comments
Author Response
Comments 1: Thank you for addressing the comments
Response 1: We deeply appreciate your review.